# SMALL FEATURES MATTER:
# ROBUST REPRESENTATION FOR WORLD MODELS

## ABSTRACT

In Model-Based Reinforcement Learning (MBRL), an agent learns to make decisions by building a world model that predicts the environment's dynamics. The accuracy of this world model is crucial for generalizability and sample efficiency. Many works rely on pixel-level reconstruction, which may focus on irrelevant, exogenous features over minor, but key information. In this work, to encourage the world model to focus on important task related information, we propose an augmentation to the world model training using a temporal prediction loss in the embedding space as an auxiliary loss. Building our method on the DreamerV3 architecture, we improve sample efficiency and stability by learning better representation for world model and policy training. We evaluate our method on the Atari100k and Distracting Control Suite benchmarks, demonstrating significant improvements in world model quality and overall MBRL performance.

## 1 INTRODUCTION

In Model-Based Reinforcement Learning (MBRL), the goal is to learn a useful policy by predicting the dynamics of the environment and rolling out the steps in the model. This model — often referred to as a world model — acts as a surrogate for the environment, allowing the agent to simulate its actions while preventing actual interaction with it. Thus, accurately modeling the dynamics of the environment in a sample-efficient manner is essential towards a sample-efficient reinforcement learning (RL) agent.

A key challenge in RL is to distinguish endogenous and exogenous information. Endogenous information is the internal, task-relevant information that directly influences the dynamics of the environment and the success of the agent's policy. Exogenous information is the irrelevant or non-essential information that does not influence the underlying dynamics of the task or the agent's policy. Pixel-level reconstruction is utilized to learn the world model by many (Hafner et al., 2020; 2021; 2023; Micheli et al., 2022; Zhang et al., 2024), however, it has pitfalls that we aim to tackle in this paper. Firstly, limited bandwidth may favor minor endogenous information over major exogenous information. Also, the RL agent might learn to rely on exogenous information for the policy, resulting in a subpar and spurious policy. Learning to represent minor exogenous information is crucial, which can be illustrated with "breakout" game from Atari. In this game, the player controls a paddle to bounce a small ball and break bricks in a big black space. Being only one pixel out of 4096, the ball is minor endogenous information, however, the black space is major exogenous feature. If we cannot represent the ball properly, the model can only learn from the movement of the paddle and the disappearance of the pixels, which is a harder task than the original. We show that by representing the ball

There are many recent works that improve RL agent performance by adding auxiliary losses in order to learn a better representation (Lamb et al.; Islam et al., 2023; Stooke et al., 2021; Yu et al., 2022; Zang et al.). While these methods generally involve a feedback loop from the RL agent that encourages learning a better representation, Stooke et al. (2021) takes a different approach. Notably, they aim to learn the representation by using only self-supervised learning (SSL) — specifically, learning a forward prediction in the latent space, where the prediction is between the features of temporally distant observations. This decoupling between the RL agent and representation learning connects well to our MBRL case — RL agent training and world model training are also decoupled. This

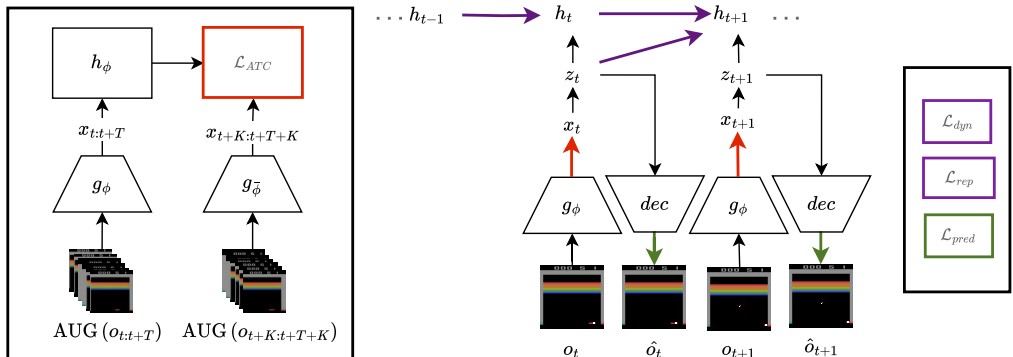

Figure 1: We integrate ATC loss $\mathcal{L}_{\text{ATC}}$ into the DreamerV3 architecture additionally to $\mathcal{L}_{dyn}$, $\mathcal{L}_{rep}$ and $\mathcal{L}_{pred}$. At first, we augment two sequences of observations — the anchor $o_{t:t+T}$ and positive $o_{t+K:t+T+K}$ — which are then encoded into latent representations $x_{t:T+T}$ and $x_{t+K:t+T+K}$ with encoders $g_\phi$ and $g_{\bar{\phi}}$. By contrasting the predictions given by anchor latents and the positive latents, we compute $\mathcal{L}_{\text{ATC}}$, encouraging the model to bring temporally distant observations closer in the latent space. The learnt encoder maps observation $o_t$ to latent representation $x_t$, which is passed to the DreamerV3's RSSM world model to predict the future latent representation $z_{t+1}$ from hidden state $h_t$ and action $a_t$. These $z_t$ observations are then decoded to reconstruct the observations $\hat{o}_t$ and $\hat{o}_{t+1}$.

similarity motivates us to pursue the applicability of the technique in learning a better representation for world models.

In this work, we tackle the following question:

*Can a forward-predictive scheme in the joint embedding space help to learn a better representation for training world models?*

There are important considerations that come with the question above. 1) The representation learnt from the scheme should focus on features that are necessary for world models — features that are useful for dynamics prediction. 2) The features should be sufficiently endogenous — useful for the RL agent to train its policy on. To investigate the question, we propose an auxiliary loss incorporated into the DreamerV3 architecture (Hafner et al., 2023). Specifically, we augment observations by applying random shifts and learn a joint-embedding representation by predicting forward steps. We empirically show that our method learns endogenous representation that is useful for the world model and, subsequently, RL policy training. Our method is the closest to the work of Stooke et al. (2021), although our work finds its purpose in the context of world models. It can also be considered as a variant of Joint Embedding Predictive Architecture (JEPA) (LeCun, 2022; Assran et al., 2023) with augmented input frames. Our primary contributions are the following.

- We introduce joint embedding forward prediction loss as an auxiliary loss for world model representation learning.

- We empirically show that our method built on DreamerV3 learns useful representation for MBRL in the Atari100k benchmark.

- We empirically show that our method learns a robust representation by evaluating on the Distracting Control Suite Benchmark.

- We study the role of forward prediction steps for sample efficiency.

## 2  RELATED WORKS

**MBRL and World models**   World models (Ha & Schmidhuber, 2018) improve sample-efficiency (Mahe et al., 2021) to learn the state transition dynamics and maximize the return (Xu et al., 2018). It has been shown both empirically (Jiang et al., 2020) and theoretically (Sun et al., 2019) that MBRL yields a more accurate policy than model-free RL. The idea of learning a model of the environment and using the world model for learning a task has been around for many years (Nguyen & Widrow, 1990; Schmidhuber, 1990; Jordan & Rumelhart, 2013). Early works in MBRL typically relied on low-dimensional state spaces for model prediction (Williams et al., 2017; Janner et al., 2019; 2020), with limited applicability in larger, complex tasks. Therefore, many recent works (Gelada et al., 2019; Rafailov et al., 2021; Hafner et al., 2023) have focused on learning models of environments that the agent can interact with to solve complex tasks. These world models also include hierarchical world models that utilize goal-conditioned RL (Gumbsch et al., 2023), hierarchical exploration (Mattes et al., 2024), and multi-time scale prediction (Shaj Kumar et al., 2023).

**Representation learning in MBRL**   Many recent efforts have been put on to the construction of precise and sample-efficient models of the environment (Hafner et al., 2020; 2021; 2019a), with some work solely focusing on constructing a robust MDP in a stricter setting (Hafner et al., 2019a; Agarwal et al., 2020). Some works rely on temporal consistency for learning the world model (Zhao et al., 2023; Hansen et al., 2022; 2024; Yan et al., 2023). The DreamerV1-3 models (Hafner et al., 2020; 2021; 2023) use RSSM (Hafner et al., 2019b) to learn a latent representation of the world and use the hidden states to predict the next abstract states. Many works build on the Dreamer architecture by augmenting it with different ideas, such as learning task informed representations coming from the policy (Fu et al., 2021), modeling two independent latent MDPs that represent useful signal and noise (Wang et al., 2022), utilizing a contrastive loss (Okada & Taniguchi, 2021; Poudel et al., 2024), using mutual information Zhu et al. (2023), and reconstruction in the latent space (Sun et al., 2024). There are multiple pathways for using a transformer as the backbone of the world models. IRIS (Micheli et al., 2022) uses VQ-VAE (Kingma, 2013; Van Den Oord et al., 2017) to quantize the observations into tokens to learn a GPT2 (Radford et al., 2019) backbone for the world model. Delta IRIS (Micheli et al., 2024) improves the efficiency by encoding the stochasticity and simulating the resulting tokens. Furthermore, STORM (Zhang et al., 2024) uses a stochastic transformer to improve the efficiency and precision of the world model. Unlike previous approaches that rely on task specific information or noise modeling, we learn representations by predicting in the latent space.

## 3  PRELIMINARIES

**Reinforcement Learning.**   In RL, an agent uses trial and error to explore an environment and improve its decision-making strategy. This agent interacts with the environment by taking actions and receiving rewards to learn an optimal policy that maximizes cumulative returns. This interaction is typically modeled as a Markov Decision Process (MDP) $\mathcal{M} = (\mathcal{S}, \mathcal{A}, P, R, \gamma)$. In the MDP, $\mathcal{S}$ denotes the state space, $\mathcal{A}$ the action space, $P(s' \mid s, a)$ the transition probability between states given an action, $R(s, a)$ the reward function, and $\gamma \in [0, 1]$ the discount factor that prioritizes immediate rewards over distant ones. The goal of RL is to learn a policy $\pi(a \mid s)$ that maximizes the expected discounted reward $J(\pi) = \mathbb{E}_\pi \left[ \sum_{t=0}^\infty \gamma^t R(s_t, a_t) \right]$ in order to solve problems.

**Model-based RL.**   In MBRL, the agent learns a world model of the environment's dynamics, represented by the transition function $\hat{P}(s' \mid s, a)$ and the reward function $\hat{R}(s, a)$. Given a current state $s_t$ and action $a_t$, the world model predicts the next state $s_{t+1} \sim \hat{P}(s_{t+1} \mid s_t, a_t)$ and the corresponding reward $r_t = \hat{R}(s_t, a_t)$. Using these predictions, the agent can simulate future trajectories by iterating over sequences of states and actions. The goal is to optimize the policy $\pi(a \mid s)$ by maximizing the expected reward over these simulated trajectories $\mathbb{E}_\pi \left[ \sum_{t=0}^T \gamma^t \hat{R}(s_t, a_t) \right]$. By using the learned world model $\hat{P}(s' \mid s, a)$ and reward $\hat{R}(s, a)$, the agent learns to refine its policy $\pi(a \mid s)$, improving performance without relying extensively on direct interaction with the real environment.

## 4 METHODS

Endogenous information for an RL agent is often similar to that needed to predict environmental dynamics. Effective transmission of key information between the dynamics and the feature extractor is crucial, as it enables the representation of relevant endogenous features while filtering out irrelevant exogenous ones. Minimizing prediction loss in a shared latent space between temporally distant observations helps the feature extractor learn more robust representations, promoting information flow between the dynamics and the features.

To enhance this information flow in MBRL, we propose augmenting world model training with joint embedding forward prediction. Specifically, we integrate the Augmented Temporal Contrast (ATC) loss as an auxiliary loss into DreamerV3, improving the representation learned by the world model and leading to a more sample-efficient policy learning. We start with a short description of DreamerV3 and then go on to describe the ATC loss and its integration into the DreamerV3 architecture. Finally, we describe the agent policy sampling with our modification. The pipeline is shown in Figure 1.

### 4.1 DREAMERV3

The DreamerV3 architecture uses a Recurrent State Space Model (RSSM) (Hafner et al., 2019b) to learn a latent dynamics representation from the visual sensory signals. It optimizes its model to imagine trajectories conditioned on actions similar to the environment steps. The policy is trained with the abstract trajectories with the representations learned by the world model. The world model uses an encoder $g_\phi$ to project the sensory observations $o_t$ to latent representation $x_t$. The architecture of this encoder is presented in Appendix D. The latent representations are transformed into stochastic representations $z_t$ using a representation model. Next, RSSM with hidden $h_t$ and past actions $a_{t-1}$ predicts the representation $\hat{z}_t$. Together with $z_t$ and $h_t$, we predict rewards $\hat{r}_t$, episode continuation flag $\hat{c}_t \in \{0, 1\}$, and the reconstruction of the sensory observation $\hat{o}_t$. In summary, the world model is a combination of the following.

$$
\begin{aligned}
&\text{Sequence model:} && h_t = f_\phi(h_{t-1}, z_{t-1}, a_{t-1}) \\
&\text{Representation model:} && z_t \sim q_\phi(z_t \mid h_t, x_t) \\
&\text{Dynamics predictor:} && \hat{z}_t \sim p_\phi(\hat{z}_t \mid h_t) \\
&\text{Reward predictor:} && \hat{r}_t \sim p_\phi(\hat{r}_t \mid h_t, z_t) \\
&\text{Continue predictor:} && \hat{c}_t \sim p_\phi(\hat{c}_t \mid h_t, z_t) \\
&\text{Decoder:} && \hat{o}_t \sim p_\phi(\hat{o}_t \mid h_t, z_t).
\end{aligned}
$$

### 4.2 MODIFYING DREAMERV3 WITH ATC LOSS

The inspiration in our method lies in the findings of Nayebi et al. (2023). This work highlights the brain's ability to anticipate future events based on adaptable visual representations, suggesting that forward prediction mechanisms are crucial to learn dynamic environments. Motivated by this, we integrate the ATC loss directly into the DreamerV3 architecture, enabling the model to learn a better representation of the dynamic settings.

The augmented temporal loss is forward prediction loss, where the encoder model $g_\phi$ learns to associate an observation $o_t$ with an observation K-step forward $o_{t+K}$ in the joint embedding space. The observations in a trajectory $\tau = (o_t, o_{t+1}, \ldots, o_{t+T}, \ldots, o_{t+T+K})$ are split into two groups: the anchor $(o_t, o_{t+1}, \ldots, o_{t+T})$ and the positive $(o_{t+K}, o_{t+1+K}, \ldots, o_{t+T+K})$ trajectories, where $t$ and $T$ denotes the start and the end point of the positive trajectory, respectively. Both the positive and anchor are mapped into a shared latent space $(x_t, x_{t+1}, \ldots, x_{t+T}, \ldots, x_{t+T+K})$. In this latent space, we use a predictor $h_\phi$ to predict the positive embeddings $(x_{t+K}, x_{t+1+K}, \ldots, x_{t+T+K})$ from the anchor embeddings $(x_t, x_{t+1}, \ldots, x_{t+T})$. Following Stooke et al. (2021) we compute the InfoNCE loss (Gutmann & Hyvärinen, 2010; Oord et al., 2018) on the predicted and positive embeddings, enabling the model to learn temporal relationships by minimizing the difference between the current and future embeddings. We regularize the model by updating a target encoder $g_{\bar{\phi}}^-$ using

exponential moving averages (EMA) (He et al., 2020) and augmenting the sensory observations with data transformations (Laskin et al., 2020; Yarats et al., 2021).

**Implementation.** The DreamerV3 model maps observation $o_t$ to stochastic representation $z_t$ through utilizing the convolution encoder and RSSM. To fit ATC into the method, we break down the two steps and label the intermediate output from the encoder $x_t$. Our goal is to learn the encoder that maps the sensory observation $o_t$ to latent representation $x_t$ in order to be passed to the RSSM world model. This, in essence, improves the representation that is used for the stochastic representation $z_t$, affecting the entire world model. Following Stooke et al. (2021), we utilize the components below in the forward predictive architecture. For clarity, we follow the notations as closely as possible. For additional details on implementation, please refer to Appendix C.

- **Convolution encoder and its target encoder** The encoders $g_\phi$ and $g_{\bar{\phi}}^-$ map the observations $o_t$ to a shared latent space, i.e., $x_t = g_\phi(\text{AUG}(o_t))$, where AUG is a shift augmentation. The encoder and its target act as a filter for the augmented anchor and positive observations, respectively. We use the same convolution encoder as used in the DreamerV3 architecture. We detail this encoder in Table 1.

- **Recurrent predictor $h_\phi$.** Similar to prior works (Pathak et al., 2017; Islam et al., 2023; Assran et al., 2023), we use a recurrent layer $h_\phi$ to carry out the forward prediction $p_t$ of the latent features $x_t$, i.e., $p_t = x_t + h_\phi(x_t)$. The skip connection is useful for propagating information throughout the model. Here, $p_t$ is expected to contain information about the latent features of the augmented forward observations.

- **Contrastive transformation matrix $W_\phi$.** The matrix W captures the information between anchors and positives by contrasting the anchor embeddings to the positive embeddings. Given the forward prediction $p_t$ and positive embedding $x_{t+K}$, the logits are $l_{t,t+K} = p_t W_\phi x_{t+K}$. The logits capture the model's belief of the relationship between the forward prediction $p_t$ and positive embedding $x_{t+K}$. We calculate the differences between the logits and true labels using a cross-entropy loss, which is the following.

$$\mathcal{L}_{\text{ATC}}(\phi) = -\log \frac{\exp l_{t,t+K}}{\sum_{t \in 0,\dots,T-1} \exp l_{t,t+K}}. \tag{1}$$

The loss $\mathcal{L}_{\text{ATC}}$ maximizes the expected agreement between the augmented anchor and forward observations in the latent space.

**Regularization using stochastic augmentation.** To regularize the training, we augment the observations with random shifts. To achieve this, we first move the images in randomly picked one of the four directions for $k_{pad}$ pixels. This creates empty pixels, which we fill with its closest pixel. Throughout our implementation, we keep $k_{pad}$ fixed at 4. We apply the shift augmentation to both anchors and positives. Notably, we only apply this augmentation to the sampled trajectories during calculating the ATC loss.

**Combining with DreamerV3 losses.** We base our implementation by incorporating the losses reported in the DreamerV3 architecture. The representation and the dynamics loss involve the KL divergence between the posterior state $z_t$ and the predicted prior state $\hat{z}_t$ with free bits (Kingma et al., 2016). Next, we train the reward predictor and decoder via the symlog loss and the continue predictor via binary classification loss. Overall, these losses are the following.

$$\mathcal{L}_{\text{pred}}(\phi) \doteq -\ln p_\phi(o_t \mid z_t, h_t) - \ln p_\phi(r_t \mid z_t, h_t) - \ln p_\phi(c_t \mid z_t, h_t) \tag{2}$$

$$\mathcal{L}_{\text{dyn}}(\phi) \doteq \max\left(1, \text{KL}\left[\text{sg}(q_\phi(z_t \mid h_t, x_t)) \,\|\, p_\phi(z_t \mid h_t)\right]\right) \tag{3}$$

$$\mathcal{L}_{\text{rep}}(\phi) \doteq \max\left(1, \text{KL}\left[q_\phi(z_t \mid h_t, x_t) \,\|\, \text{sg}(p_\phi(z_t \mid h_t))\right]\right) \tag{4}$$

Putting it all together, the total loss is:

$$\mathcal{L}(\phi) \doteq \mathbb{E}_{q_\phi}\left[\sum_{t=1}^{T} \left(\beta_{\text{pred}}\mathcal{L}_{\text{pred}}(\phi) + \beta_{\text{dyn}}\mathcal{L}_{\text{dyn}}(\phi) + \beta_{\text{rep}}\mathcal{L}_{\text{rep}}(\phi) + \beta_{\text{ATC}}\mathcal{L}_{\text{ATC}}(\phi)\right)\right]. \tag{5}$$

We train the parameters of the different components of the world model together with this loss.

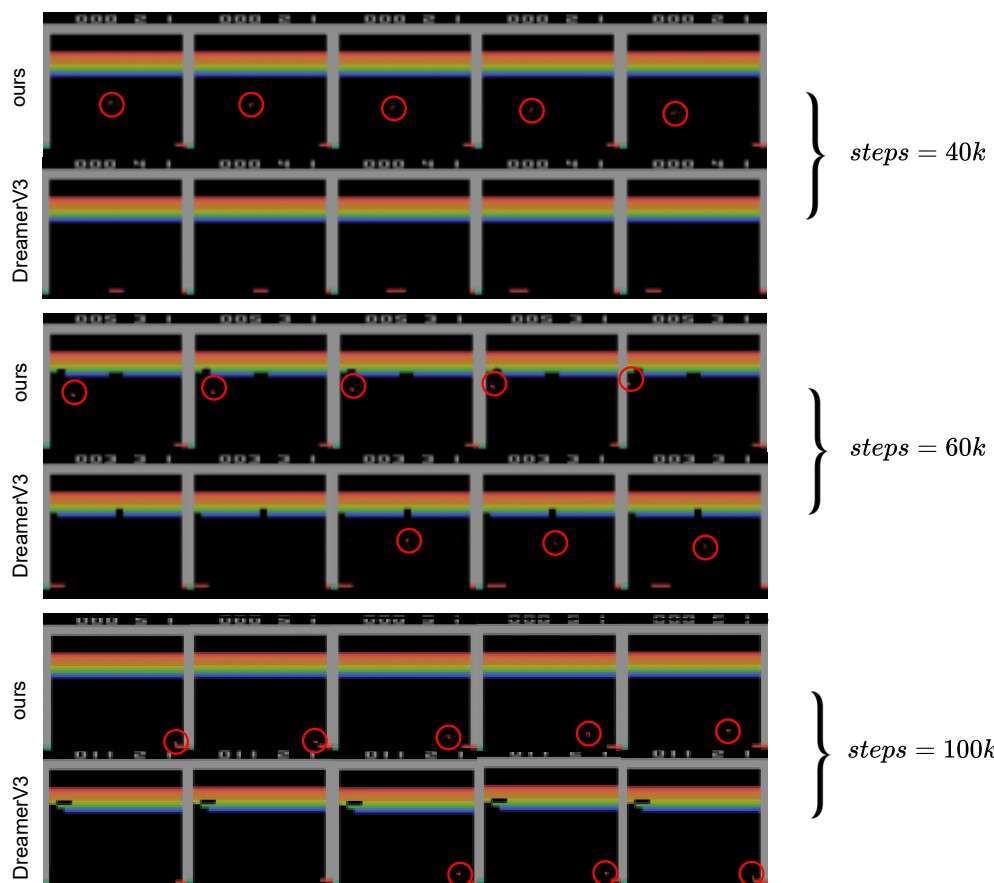

Figure 2: Comparison of representations learned by our model with those generated by DreamerV3 (both rollout). In all 40k, 60k and 100k iterations, our method consistently represents the ball properly. However, DreamerV3 does not captures this crucial detail at 40k iterations. In 60k and 100k iterations the ball disappears in the representation of DreamerV3 after a briefly appearance. Unlike DreamerV3, which struggles to consistently capture key information, our model remains robust in visualizing the ball throughout.

### 4.3 THE POLICY

We follow the actor-critic setup from the DreamerV3 — a short summary of which can be found in the Appendix B. To learn the policy, we roll out trajectories in the abstract space and train the actor-critic model on them. In order to sample training trajectories from the policy to learn the world model, we use the target encoder $g_{\bar{\phi}}$ to encode the sensory observations.

## 5 EXPERIMENTS

We present our experimental results on the Atari100k benchmark with mean return on the evalutation stage. Empirically, we aim to address the following questions.

1. Does our method learn useful endogenous representation for the world model?

2. Are the representations learned by the world model useful in downstream tasks, e.g., learning a better RL policy?

3. How does our method behave in the presence of exogenous information?

4. How do different environment dynamics affect the optimal forward prediction horizon K?

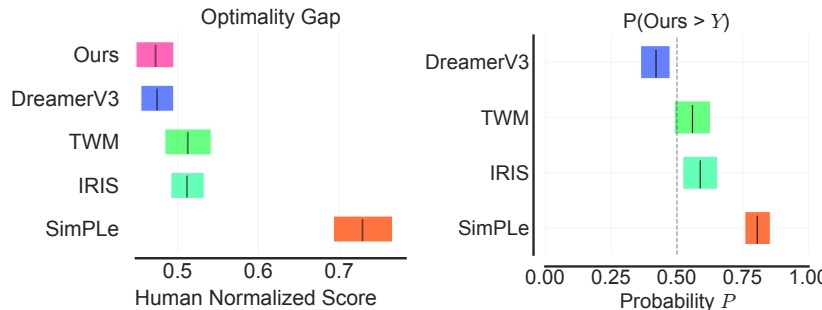

Figure 3: Comparison of our method and baselines across two metrics. (Left) The Optimality Gap shows how much each algorithm falls short of human-level performance (smaller is better). Our method exhibits the smallest gap, indicating that it comes closest to replicate human-level performance outperforming baselines. (Right) The Probability of Improvement highlights the likelihood of our method outperforming the baselines on a random task. We observe a high probability of improvement over all baselines, with the exception of DreamerV3 (with reported scores), where the margin is narrower.

We address the questions above in the next part of the paper.

### 5.1 ATARI100K BENCHMARK

The Atari 100k benchmark is used to test RL algorithms using only 100,000 environment steps. It includes a subset of 26 Atari games and focuses on data efficiency, requiring agents to perform well with limited training data. This benchmark is widely used to evaluate the sample efficiency and generalization capabilities of RL models.

**Baselines.** We compare the following methods as a baseline. SimPLe (Kaiser et al., 2019) trains a policy using PPO (Schulman et al., 2017) leveraging a world model represented as an action-conditioned video generation model; TWM (Robine et al., 2023) uses a transformer-based world model that leverages a Transformer-XL architecture and a replay buffer that uses a balanced sampling scheme (Dai et al., 2019); IRIS (Micheli et al., 2022) uses a VideoGPT Yan et al. (2021) based world model; DreamerV3 (Hafner et al., 2023), a general algorithm that achieves SOTA results on a multitude of RL benchmarks.

**Metrics.** RL evaluation is difficult due to stochasticity and computational complexity related to environments (Agarwal et al., 2021). Keeping this in mind, we provide a series of related metrics to evaluate the overall performance of our method on the Atari100k benchmark. The metrics are the following: human mean, human median, Interquantile Mean (IQM), optimality gap, performance profiles, and probability of improvement. We provide a full description of these metrics in Appendix E.

**Results.** We report our results in Figure 3 and complete benchmark results in Appendix F. Our method achieves the best result over 11 games while outperforming or equaling DreamerV3 in 18 games. We present an optimality gap of 0.473 smaller than the DreamerV3 baseline. Additionally, we report a high probability of improvement over baselines on a random task except for DreamerV3, shown in Figure 3. Finally, we achieve significantly better results in human normalized mean, median, and IQM than other baselines.

### 5.2 EVALUATING THE LEARNED REPRESENTATION

Whether the representations learned by our world model are mappable to useful sensory observations is a good indication of the quality of the representations. This is also convenient for us, since we already train a decoder using reconstruction loss. With this in mind, we examine the reconstructed rollout frames from DreamerV3 and our method at 40k and 60k steps for Breakout. Figure 2 shows the results. At 40k steps, the reconstructed observation from DreamerV3 cannot model the red ball, while our method faintly models the ball. Furthermore, at 60k, we can see that both methods

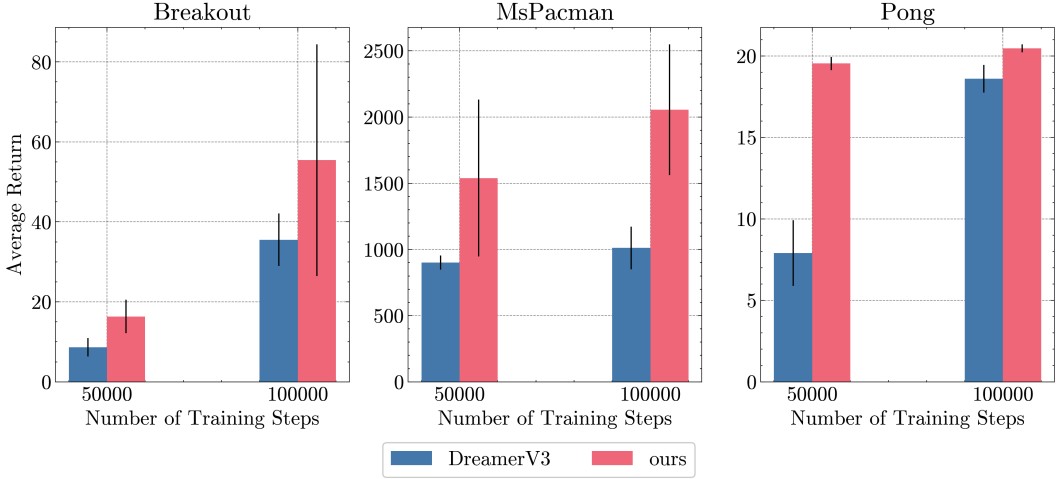

Figure 4: Performance comparison of our proposed method (red) and DreamerV3 (blue) in three Atari environments: Breakout (left), Ms. Pac-Man (middle), and Pong (right). In all environments, our method not only outperforms DreamerV3, but demonstrates a faster growth overall. We achieve strong results quickly (50k), which contributes to the overall high final performance.

clearly model the ball, but DreamerV3 misses the ball at a few time steps. Since the red ball is an important learning signal for the RL agent as well as environment dynamics, this experiment shows the improved quality of the representations learned by our method.

## 5.3 EVALUATING THE USEFULNESS OF THE REPRESENTATIONS IN DOWNSTREAM TASKS

Since the representation learned by the world model is completely decoupled from the actor-critic agent, another indication for a good representation is whether it leads to a better RL policy. Hence, if the world model is able to capture endogenous information quicker, it should also reflect in the RL agent's policy, given the task in hand. To test our hypothesis, we compare the evaluation returns between the RL agents of DreamerV3 and our method in Breakout, Ms. Pac-Man, and Pong across two stages of the training. We report the results in Figure 4. The results make the distinction clear that our method learns endogenous representations quicker than the DreamerV3, leading to a more sample-efficient RL policy.

## 5.4 EVALUATING THE ROBUSTNESS OF THE REPRESENTATIONS

To evaluate our method's robustness against different exogenous information, we perform two experiments. First, we add pixel-wise noise to selected Atari games and compare our method against its baseline, DreamerV3. Next, we take on the Distracting Control Suite Stone et al. (2021) benchmark, which adds dynamic backgrounds to the DeepMind Control Suite Tassa et al. (2018) tasks.

**Injection of pixel-wise noise.** For this task, we inject pixels with noise during training the world model. We control the difficulty of the task by adjusting the pixel shift probability $p_{shift}$, where higher $p_{shift}$ corresponds to a higher task difficulty. The representation learner needs to be robust to noise to filter out the exogenous information passed to the downstream RL agent. To evaluate the robustness of our method, we compare it against DreamerV3 on two difficulties of the task, $p_{shift} \in \{0.01, 0.05\}$ over 3 Atari games: Asterix, Alien, and Breakout. We report the results in Figure 5. We see that with increasing difficulty of the tasks, our method keeps its result consistent, while DreamerV3 does not.

**Dynamic background as an exogenous information.** The Distracting Control Suite provides challenging tasks with exogenous information for world model evaluation. The tasks are divided into three difficulties: easy, medium, and hard. We compare our method with DreamerV3 on the Cheetah run task over three difficulties: easy, medium, and hard. Figure 10 in Appendix G shows the

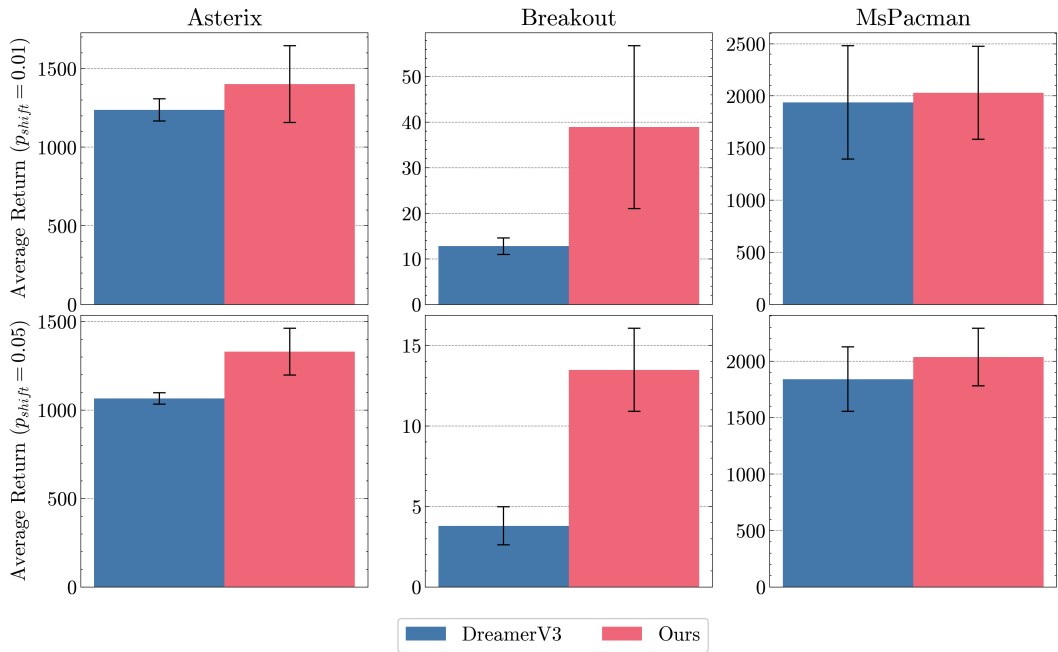

Figure 5: Average evaluation return of our method compared to DreamerV3 at 100k steps. We present results across three environments: Asterix (left), Breakout (middle), and Ms. Pac-Man (right). In order to test robustness, we experiment with noise injection to the data using a probability of $p_{shift} = 0.01$ *(top)* and $0.05$ *(bottom)*. With increasing difficulty, our method consistently gets good scores in all three environments, while the score of DreamerV3 drops, showing that by learning a better representation of the data, our model is more robust than DreamerV3.

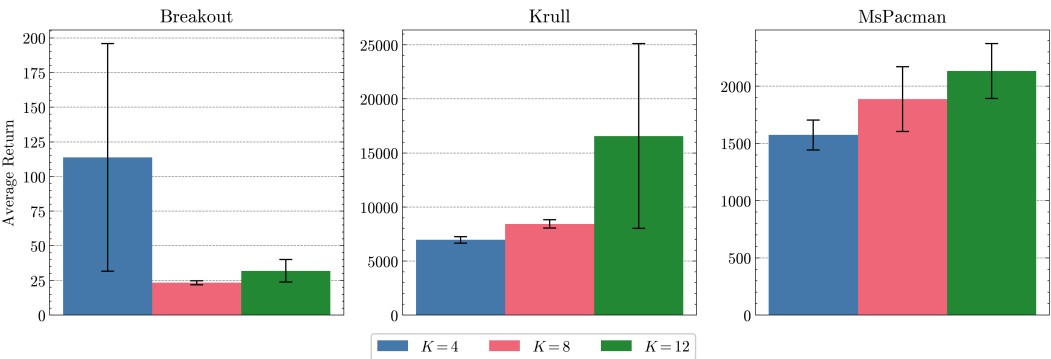

Figure 6: Average evaluation return of our method at 100k iterations using $K \in \{4, 8, 12\}$ across three Atari100k environments: Breakout (left), Krull (middle), and Ms. Pac-Man (right). The results show that environments with complex dynamics such as Krull and Ms. Pac-Man and significant input distribution shifts perform better with higher values of $K$, while simpler games like Breakout benefit from lower values.

results. For the easy task, we see that the methods perform comparably. However, as the difficulty increases, our method stays robust to the dynamic background, resulting in a slighter drop in the scores compared to DreamerV3. For a detailed discussion on what affects the task difficulty, please see Appendix G.

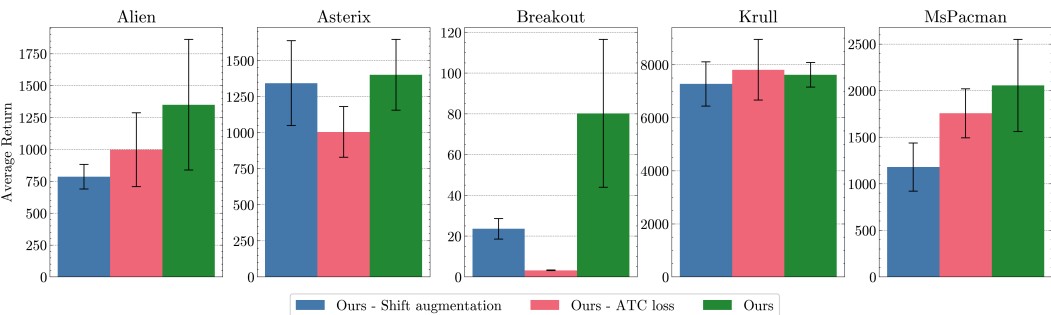

Figure 7: Ablation on ATC loss and shift augmentation in five Atari games: Alien, Asterix, Breakout, Krull, Ms-PacMan. The results highlight that ATC loss and shift augmentation combined boosts the performance in four environments, while in Krull, the effect of these components appears negligible.

## 5.5 Evaluating the effect of K

To evaluate the effect of the hyperparameter $K$, we compare our method for $K \in \{4, 8, 12\}$. We pick three games for this evaluation: Breakout, Krull, and Ms. Pac-Man. Figure 6 reports our findings. We see two different phenomenon: games with complex dynamics such as Krull and Ms. Pac-Man favor higher values of $K$, whereas simple games such as Breakout favor smaller values of $K$. This finding is also consistent with the ones from Mattes et al. (2024), where the authors observe the importance of increasing hierarchies in complex games. However, despite the importance of $K$, we keep $K$ fixed at 4 for all experiments to keep consistency.

## 5.6 Ablation studies on ATC loss and shift augmentation

To evaluate the individual contribution of the shift augmentation and ATC loss of our method, we perform an ablation study on three Atari games (Alien, Asterix, Breakout, Krull, Ms-PacMan). Figure 7 illustrates the results, showing that both ATC loss and the shift augmentation has substantial impact in environments such as Alien, Asterix, Breakout, Ms-PacMan. However, its influence appears negligible in Krull, although notably, our method improves it substantially with a higher value of $K$ (see Section 5.5).

## 6 Discussions and conclusions

In this work, we propose an augmentation to the world model representation learner by using forward prediction loss in the joint embedding space. Using our augmentation for world model training, we achieve a more robust world model, which helps downstream MBRL tasks. We show strong empirical results in the Atari100k benchmark, especially when the frames are injected with noise. We also show our method's robustness to exogenous features with strong empirical results in the Distracting Control Suite benchmark.

While our results are promising, the primary limitation of our study lies in the considered benchmarks. Due to computational limitations, we evaluated our methods for all experiments in three seeds, considered fewer steps, and performed evaluations less frequently. In addition, while our method learns task-relevant features, it is not a task-specific learner. We envision a task-specific feature learner as a two-part problem: a causal classifier of task-specific features and a feature extractor given those features. We leave this task for future work.

Despite these limitations, our method enhances world model sequence modeling capabilities and improves downstream tasks by employing a temporal prediction loss in the joint embedding space as an auxiliary loss. Indeed, as discussed in section 4, having such a loss enforces the feature extractor to learn a temporally relevant representation by sharing information between it and the dynamics. This information passing improves downstream tasks by filtering out exogenous features. For example, in Figure 2, our method can represent the ball because being able to do so is important to predict the state $K$ steps later. Without enforcing this, we see that DreamerV3 fails to represent it consistently.

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

## A  HYPERPARAMETERS

We present all hyperparameters in this section.

| | | |
|---|---|---|
| Training batch size | $B$ | 16 |
| Training batch length | $T$ | 64 |
| Imagination horizon | $L$ | 15 |
| Gamma | $\gamma$ | 0.997 |
| Lambda | $\lambda$ | 0.95 |
| Entropy Coefficient | $\eta$ | $3 \times 10^{-4}$ |
| Optimizer | - | Adam |
| World model learning rate | - | $1 \times 10^{-4}$ |
| World model gradient clipping | - | 1000 |
| Actor-critic learning rate | - | $3.0 \times 10^{-5}$ |
| Actor-critic gradient clipping | - | 100 |
| Gray scale input | - | False |
| Frame stacking | - | False |
| Dynamics hidden dimension | - | 512 |
| Dynamics deterministic dimension | - | 512 |
| Dynamics stochastic dimension | - | 32 |
| Dynamics discrete dimension | - | 32 |
| Dynamics activation | - | Sigmoid |
| Normalization | - | True |
| Encoder-decoder MLP layers | - | 5 |
| Encoder-decoder MLP dimension | - | 1024 |
| Encoder-decoder CNN depth | - | 32 |
| Encoder-decoder CNN dimension | - | 1024 |
| Encoder-decoder CNN bottleneck | - | 256 |
| Encoder-decoder activation function | - | SiLU |
| Actor-critic layers | - | 2 |
| Actor distribution | - | Normal |
| Critic distribution | - | Symlog (Hafner et al., 2023) |
| Forward prediction step | K | 4 |
| Dynamics loss scale | - | 0.5 |
| Representation loss scale | - | 0.1 |
| Weight decay | - | 0.0 |

## B  ACTOR-CRITIC POLICY

For our work, we adopt the existing actor-critic architecture from DreamerV3 (Hafner et al., 2023), where we train our actor-critic policy with the following reinforce estimator (Williams, 1992) loss:

$$L(\theta) := -\sum_{t=1}^{T} \mathrm{sg}\left( (\hat{R}_t - \psi(s_t)) / \max(1, S) \right) \log \pi_\theta(a_t|s_t) + \eta H[\pi_\theta(a_t|s_t)]. \tag{6}$$

We also normalize the returns by computing the range from the 5th to the 95th return percentile over the return batch and smooth out the estimate using EMA:

$$S := \mathrm{EMA}(\mathrm{Per}(\hat{R}_t, 95) - \mathrm{Per}(\hat{R}_t, 5), 0.99). \tag{7}$$

## C  DETAILS OF IMPLEMENTATION

**Shift augmentation.**  We perform shift augmentation by randomly shifting the $64 \times 64$ pixels to one of the sides by a stride of 4. Next, we fill the pixel gaps with its neighboring pixels.

**Encoder.**  We implement the encoder with convolution layers with a final depth of 32 and a resolution of 4x4. Finally, the outputs of of the layer are passed through an MLP that projects them to a 256-dimensional space.

**Recurrent predictor.**  The recurrent predictor takes the anchor latent vectors and passes them through an MLP skip connection. The MLP is a 2-layer MLP with 512 hidden dimensions.

**Contrastive transformation matrix.**  The contrastive matrix, is a $256 \times 256$ matrix which takes the anchor and positive latents and outputs a $B \times B$ matrix, where $B$ is the batch size.

## D  DETAILED ENCODER ARCHITECTURE

We present the architecture of the encoders $g_\phi$ and $g_{\bar{\phi}}^-$ in this section.

Table 1: Structure of encoder $g_\phi$ used for forward prediction. The size of the modules is omitted and can be derived from the shape of the tensors. Conv denotes CNN layers LeCun et al. (1989) characterized by kernel = 4, stride = 2 and padding = 1. SiLU means Sigmoid Linear Unit activation functions, while LayerNorm corresponds to layer normalizations Ba et al. (2016). Flatten is employed to alter the indexing method of the tensor, while preserving the data and their original order.

| Submodule | Output tensor shape |
|---|---|
| Input image ($o_t$) | $64 \times 64 \times 3$ |
| Conv1 + LayerNorm1 + SiLU | $32 \times 32 \times 32$ |
| Conv2 + LayerNorm2 + SiLU | $64 \times 16 \times 16$ |
| Conv3 + LayerNorm3 + SiLU | $128 \times 8 \times 8$ |
| Conv4 + LayerNorm4 + SiLU | $256 \times 4 \times 4$ |
| Flatten | 4096 |

## E  ATARI100K BENCHMARK

Containing 26 games, the Atari100k benchmark is a data-efficiency benchmark designed to test the performance of the RL model in 100k environment steps. It pushes models to learn more efficiently with fewer interactions, reflecting real-world constraints. Many state-of-the-art models have been tested on Atari100k (Hafner et al., 2023; Micheli et al., 2022; Fu et al., 2021), evaluating their ability to balance between performance and efficiency. Additionally, the benchmark presents human mean, human median, Interquantile Mean (IQM), optimality gap scores, probability of improvement and performance profiles.

**Human normalized score.**  The human normalized score compares the agent's scores to a human and a random agent, highlighting the differences between the algorithm's performance and human-level play in a specific environment. It can be calculated with the following formula:

$$\frac{\texttt{agent\_score} - \texttt{random\_score}}{\texttt{human\_score} - \texttt{random\_score}}$$

**Human Mean.**  The human mean aggregates the human normalised scores for all environments. It is a general indication of an algorithm's performance relative to human benchmarks.

**Human Median.**  The human median is an aggregate metric of the human normalized scores as well as the human mean, but is insensitive to high-score environments skewing it. It's a crucial

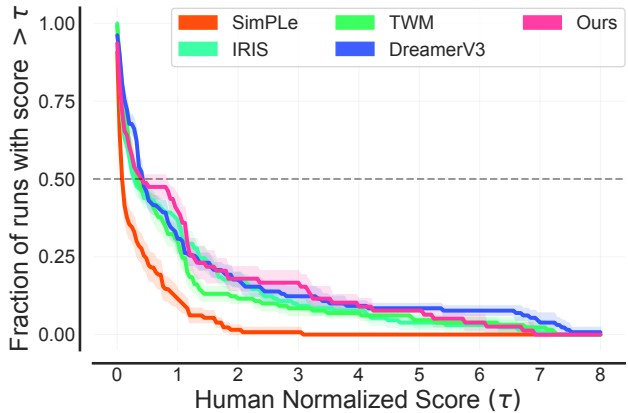

Figure 8: Performance profiles of baseline methods and our method, illustrating the fraction of games scoring higher than the specified human normalized score.

metric as it gives a balanced view of algorithm performance, particularly when some environment scores are distorting the significance of the mean.

**Interquantile Mean (IQM).** IQM is a robust statistical method excluding the top and bottom 25% of results, focusing only on the middle 50%. Mitigating the impact of outliers, it addresses the weakness of the human mean being skewed by extreme values. Considering a broader range of mid-performing environments, it is also more informative than the Human Median, which only reflects a single middle value. IQM is defined by the following formula: $\text{IQM} = \frac{1}{\lceil 0.5 \times N \rceil} \sum_{i=\lceil 0.25 \times N \rceil}^{\lceil 0.75 \times N \rceil} \text{score}_i$

**Optimality Gap.** The Optimality Gap measures the difference between the optimal human-level performance set at $\gamma = 1.0$ and the score of the algorithm. When the model consistently achieves or surpasses the target score, the optimality gap decreases. This decrease represents a strong general performance across all environments, rather than excelling in a subset of them. The optimality gap can be obtained from the following formula: optimality gap $= \max(0, 1 - \text{normalized score})$

**Probability of improvement.** The Probability of Improvement gives a probability score of how likely is algorithm $X$ better than algorithm $Y$ in a specific environment.

**Performance profiles.** Performance profiles offer a more comprehensive view of an algorithm's performance, particularly in environments, where score distributions can vary widely and have outliers. Unlike point or interval estimates — such as the human mean and human median — performance profiles capture the variability across tasks more effectively, providing deeper understanding into performance trends that single estimates might overlook.

# F    ADDITIONAL ATARI100K BENCHMARK RESULTS

In this section, we present the full evaluation and the reproduced scores of the Atari100k benchmark in Table 2.

We illustrate the optimality gap and the probability of improvement in Figure 3. The optimality gap of our method is the smallest among all, outperforming our baselines. SimPLe obtains the largest gap (0.729), significantly falling behind other methods. The gaps for TWM, IRIS, and the reported value for DreamerV3 are all above 0.5, with values of 0.513, 0.512, and 0.503, respectively. Our method is the only one below 0.5 — 0.473 — ranking first. Observing the probability of improvement scores, our method is better than DreamerV3 (reported), TWM, IRIS and SimPLe with a probability of 0.44, 0.55, 0.6, and 0.81, respectively.

In Figure 8, we compare the human normalized scores across our method and baselines.

Table 2: Evaluation on the 26 games in the Atari 100k benchmark. Following the conventions of Hafner et al. (2021), scores that are the highest or within 5% of the highest score are highlighted in bold. We highlight the scores higher than DreamerV3 with an asterisk (*).

| Game | Rand | Hum | SimPLe reported | TWM reported | IRIS reported | DreamerV3 reproduced | Ours |
|------|------|-----|---------|-----|------|-----------|------|
| Alien | 228 | 7128 | 617 | 675 | 420 | 804 | **1348*** |
| Amidar | 6 | 1720 | 74 | 122 | **143** | 122 | 128* |
| Assault | 222 | 742 | 527 | 683 | **1524** | 642 | 737* |
| Asterix | 210 | 8503 | 1128 | 1116 | 854 | 1190 | **1401*** |
| Bank Heist | 14 | 753 | 34 | 467 | 53 | 752 | **981*** |
| Battle Zone | 2360 | 37188 | 4031 | 5068 | **13074** | 11600 | 9289 |
| Boxing | 0 | 12 | 8 | **78** | 70 | 71 | **76*** |
| Breakout | 2 | 30 | 16 | 20 | **84** | 24 | 52* |
| Chopper Command | 811 | 7388 | 979 | 1697 | 1565 | 680 | 726* |
| Crazy Climber | 10780 | 35829 | 62584 | 71820 | 59234 | **86000** | **89040*** |
| Demon Attack | 152 | 1971 | 208 | 350 | **2034** | 203 | 155 |
| Freeway | 0 | 30 | 17 | 24 | **31** | 0 | 0 |
| Frostbite | 65 | 4335 | 237 | **1476** | 259 | 1124 | 1361* |
| Gopher | 258 | 2413 | 597 | 1675 | 2236 | **4358** | 3495 |
| Hero | 1027 | 30826 | 2657 | 7254 | 7037 | **12070** | 7019 |
| Jamesbond | 29 | 303 | 101 | 362 | **463** | 290 | 425* |
| Kangaroo | 52 | 3035 | 51 | 1240 | 838 | 4080 | **5227*** |
| Krull | 1598 | 2666 | 2204 | 6349 | 6616 | **7326** | **7618*** |
| Kung Fu Master | 256 | 22736 | 14862 | 24555 | 21760 | 19100 | **26744*** |
| Ms Pacman | 307 | 6952 | 1480 | 1588 | 999 | 1370 | **2056*** |
| Pong | -21 | 15 | 13 | 19 | 15 | 19 | **21*** |
| Private Eye | 25 | 69571 | 35 | 87 | 100 | **140** | 100 |
| Qbert | 164 | 13455 | 1289 | **3331** | 746 | 1875 | 1053 |
| Road Runner | 12 | 7845 | 5641 | 9109 | 9615 | **14613** | 9721 |
| Seaquest | 68 | 42055 | 683 | **774** | 661 | 571 | 547 |
| Up N Down | 533 | 11693 | 3350 | 15982 | 3546 | 7274 | **19302*** |
| Human Mean (↑) | 0% | 100% | 33% | 96% | 105% | 104% | **121%** |
| Human Median (↑) | 0% | 100% | 13% | 51% | 29% | 49% | **63%** |
| IQM (↑) | 0.00 | 1.00 | 0.130 | 0.459 | 0.501 | 0.502 | **0.589** |
| Optimality Gap (↓) | 1.00 | 0.00 | 0.729 | 0.513 | 0.512 | 0.503 | **0.473** |

## G DISTRACTING CONTROL SUITE BENCHMARK

The Distracting Control Suite Benchmark (Stone et al., 2021) is designed to test the robustness of an RL agent under visual distractions. These distractions are changes in camera poses, object colors, and background scenes, controlled by a scaler. These scalers are used to determine the difficulty of the task. The easy scenarios have minimal variation in the factors, while medium introduces more noticeable changes. In the hard samples, significant disruptions can be observed in the data — aggressive color changes, dynamic backgrounds. Figure 9 shows the training frames of a Cheetah run in three different difficulties.

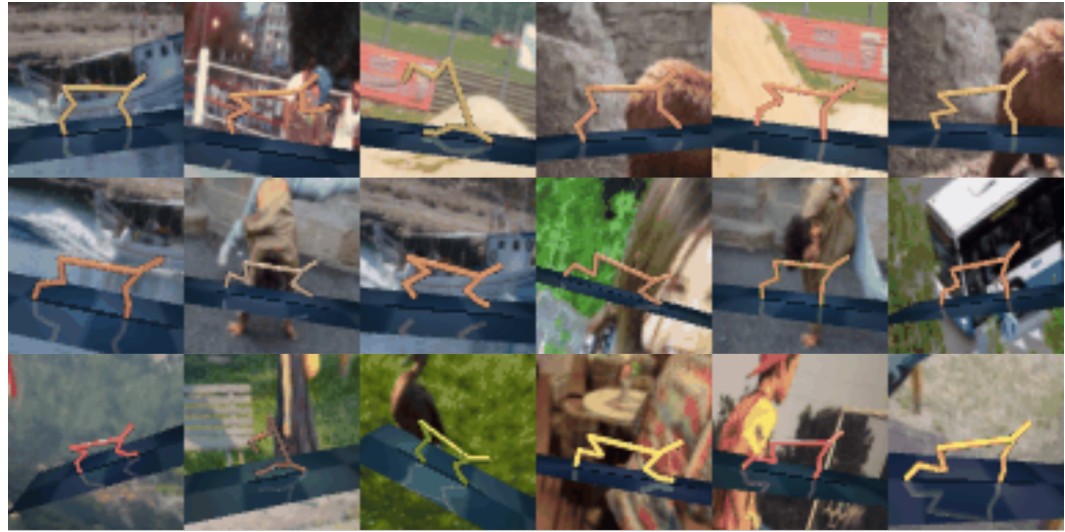

Figure 9: Training frames for the Distracting Control Suite benchmark. The frames capture the Cheetah run task across three difficulty levels: easy (first row), medium (second row), and hard (third row). We evaluate our method under varying amounts of visual disturbances using this benchmark.

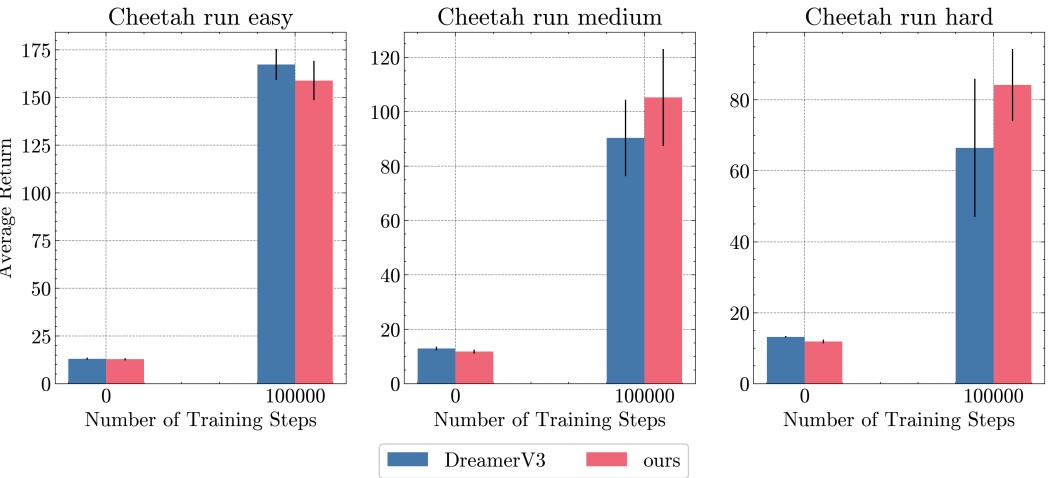

Figure 10: Performance comparison of our proposed method (red) and DreamerV3 (blue) in three increasing difficulty of Cheetah Run task in the Distracting Control Suite benchmark. Our method shows better performance than the baseline with increasing difficulty.

## H    ROBUSTNESS EXPERIMENT FRAMES

Figure 11 presents the training frames after noise injection using probabilities of $p_{shift} = 0.01$ and $p_{shift} = 0.05$. When using $p_{shift} = 0.05$ the frames are more disturbed — the extra (red) pixels make the representation of the red ball even harder.

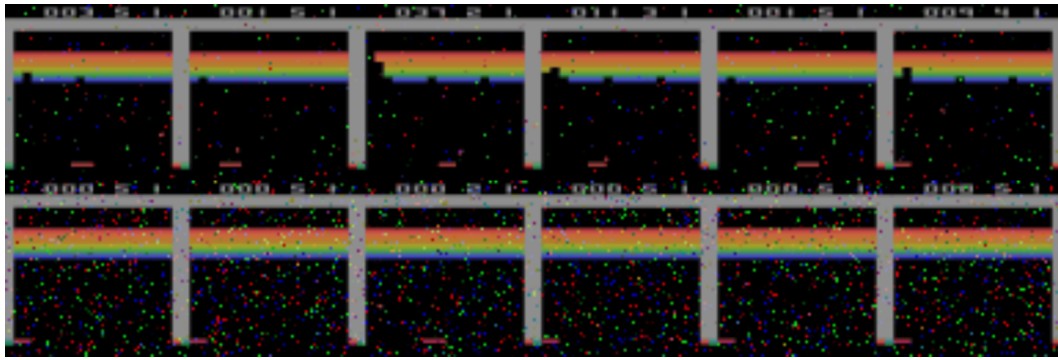

Figure 11: Training frames for the robustness experiments in the Atari Breakout environment. The top row shows $p_{shift} = 0.01$, corresponding to a lower noise level, while the bottom row represents $p_{shift} = 0.05$, indicating higher difficulty.

