# OpenReview forum: "Small features matter: Robust representation for world models"
_ICLR.cc/2025/Conference — Submitted to ICLR 2025_

### Official Review · Reviewer_mApk · 2024-10-25

**Soundness:** 2
**Presentation:** 3
**Contribution:** 2
**Rating:** 3
**Confidence:** 4

**Summary:**

The paper aims to improve the robustness of the world model training in MBRL, which is important to its generalizability and sample efficiency. Going beyond relying on pixel-level reconstruction, this paper enables the world model focusing on task-related information by augmenting world model training with a temporal prediction loss in embedding space. The experiments demonstrate its improvements in representation quality and overall performance.

**Strengths:**

1.This paper is well written, easy to follow, and the  motivation is clear.  For instance, the notion of endogenous and exogenous information is well explained with a specific example in the introduction.

2.The proposed method, though simple, makes sense, as it adds a sensible auxiliary loss function in DreamerV3 to learn a more robust representation for embedding forward prediction. The experiments demonstrate its performance.

**Weaknesses:**

1. The proposed idea is straightforward.

2. Loss clarification. The paper does not clarify  how the forward-predictive scheme with ATC loss helps to learn sufficiently endogenous features. In section 4.2 where the design of ATC loss is presented,  the relationship between these loss functions and endogenous information is not clear.  In particular, the paper should try best to explain why this loss function “prefers” endogenous information. I suggest the authors provide a theoretical justification or empirical analysis showing how the loss function encourages focus on task-relevant information.

3.	Augmentation. More details on augmentation are needed to understand the key steps in the proposed method. How does the augmentation by random shifts on observation produce the time sequences of anchor and positive from t to t+T+K? Does this augmentation involve random actions in  the environment or just random shifts that change the observation frames while the environment stays the same? It will help to provide a step-by-step description of the augmentation process.

4.	Figures. Some figures in the paper should be improved to make them more readable. For example, the ball in Figure.2 is too small to read.  Also, there is no label in the figure (only in the caption) to indicate the model of each line, which makes the figures hard to understand.  I recommend enlarging the ball in Figure 2 and providing a legend for clarity.

**Questions:**

Figure 2 shows the reconstruction in 40k and 60k steps, while the experiments are in 100k steps. How is the reconstruction of DreamerV3 in 100k steps or more? The ball is still invisible sometimes?

Can the authors compare the proposed methods against other methods, such as image augmentation, in terms of robustness? More experiments along this line will strengthen the contributions.

---

> ### Author Response · Authors · 2024-12-02
> **W1 and W2**
>
> W1. The proposed idea is straightforward.
>
> A1. Indeed, it is. In fact, our idea is straightforward purposefully.
>
> Our work is inspired by animal cognitive process, specifically from this work [1] where the authors show that animal brains' capability of processing important information mimics more closely to predicting futures in the latent representation.
>
> The authors report, "...future prediction within the latent space of a video foundation model, pretrained on diverse egocentric sources, aligns best with high-throughput neural and behavioral patterns in scenes that it was not originally trained on. Our findings indicate that primate mental simulation harbors robust inductive biases, and is so far most consistent with predicting the future state of its environment in a latent space that is reusable across dynamic environments."
>
> Our work also incorporates the same pathway. The augmented temporal contrast loss focuses on features that are useful for predicting the dynamics, which is mostly the cases with endogenous features.
>
> Motivated by [1], we note in our introduction "Can a forward-predictive scheme in the joint embedding space help to learn a better representation for training world models?". Thus, we decide to make as little as possible to the DreamerV3 to evaluate the findings in the case of world models, making our work **as simple as possible**.
>
> W2. Loss clarification. The paper does not clarify how the forward-predictive scheme with ATC loss helps to learn sufficiently endogenous features. In section 4.2 where the design of ATC loss is presented, the relationship between these loss functions and endogenous information is not clear. In particular, the paper should try best to explain why this loss function “prefers” endogenous information. I suggest the authors provide a theoretical justification or empirical analysis showing how the loss function encourages focus on task-relevant information.
>
> A2. We go into details in A1. To recap, the augmented temporal contrast loss focuses on features that are useful for predicting the dynamics, which is mostly the cases with endogenous features.
>
> We believe that the idea of temporal contrast loss focusing on dynamics-focused features, leading to endogenous features should be straightforward enough. However, we are open to ideas by the reviewers on how to empirically show the same.

---

> ### Author Response · Authors · 2024-12-02
> **W3 and W4**
>
> W3. Augmentation. More details on augmentation are needed to understand the key steps in the proposed method. How does the augmentation by random shifts on observation produce the time sequences of anchor and positive from t to t+T+K? Does this augmentation involve random actions in the environment or just random shifts that change the observation frames while the environment stays the same? It will help to provide a step-by-step description of the augmentation process.
>
> A3. Our methods apply random shifts by simply 'moving' the observation in one of the randomly chosen four directions (up, down, left, right) and filling the pixels with the neighboring pixels.
>
> It does not involve random actions in the environment. All the environment actions come from the policy. Indeed, the second one is correct, i.e., "just random shifts that change the observation frames while the environment stays the same".
>
> > It will help to provide a step-by-step description of the augmentation process.
>
> We thank the reviewer for the suggestions for improving our writing. We have provided a step-by-step description of the augmentation process on the line 248-253.
>
> W4.  Figures. Some figures in the paper should be improved to make them more readable. For example, the ball in Figure.2 is too small to read. Also, there is no label in the figure (only in the caption) to indicate the model of each line, which makes the figures hard to understand. I recommend enlarging the ball in Figure 2 and providing a legend for clarity.
>
> A4. We thank the reviewer for such detailed-oriented questions. Based on their response we have updated our figure 2 to make sure the ball is clear by adding red circles around the ball, when it is visible.
>
> We have also improved the figure by adding legends for different models at different stages of the training.
>
> We thank the reviewer and invite them to check our updated manuscript!

---

> ### Author Response · Authors · 2024-12-02
> **Q1 and Q2**
>
> Q1. Figure 2 shows the reconstruction in 40k and 60k steps, while the experiments are in 100k steps. How is the reconstruction of DreamerV3 in 100k steps or more? The ball is still invisible sometimes?
>
> A1. We thank the reviewer for such insightful question.
>
> To address the reviewer's concern, we have updated the Figure 2 with the reconstruction plots for 100k steps. Indeed, we find that while DreamerV3 is able to model the ball, it drops the balls in some subsequent frames.
>
> Q2. Can the authors compare the proposed methods against other methods, such as image augmentation, in terms of robustness? More experiments along this line will strengthen the contributions.
>
> A2. Indeed. We agree with the reviewer and we have worked on an experiment that compares our method with the following methods:
>
> Image augmentation with random shifts: We remove the ATC loss and add data augmentation to the training trajectories for the DreamerV3 world model loss. Specially, we take the observations, add random shifts to them, and use these observations to learn a world model. Indeed, these augmentations only happen during the training and not during evaluations.
>
> Augmented temporal contrast loss: We remove the data augmentations that was being applied to anchor and positive. Afterwards, we calculate the ATC loss.
>
> We compare these two scenarios with our method to understand the exact contribution of each component to the reported performance improvements.
>
> We validate the ablation on 5 Atari games (Alien, Asterix, Krull, Breakout, MsPacman) over 5 seeds. We report the results of the ablations in Section 5.6.
>
> Figure 7 illustrates that the removal of the ATC loss results in poorer performance, specially in Breakout and Asterix. Additionally, leveraging the shift augmentation has a steady impact in all the games tested. However, their influence appears negligible in Krull, which is consistent with our findings in Section 5.5 where we show it is improved with higher values of $K$.
>
> Please let us know if you have any remaining doubts or concerns.

---

### Official Review · Reviewer_cF8L · 2024-10-31

**Soundness:** 2
**Presentation:** 2
**Contribution:** 1
**Rating:** 3
**Confidence:** 4

**Summary:**

This work focuses on improving the world models of reconstruction-based MBRL approaches. It tries to address the issue of learning task-irrelevant features arising from pixel-level reconstructions by modifying DreamerV3 with an additional Augmented Temporal Contrasted (ATC) loss term. Empirically, the authors show that this results in superior sample efficiency on the Atari 100k benchmark and also give a few examples of improved representation learning.

**Strengths:**

1. The fundamental limitation of reconstruction-based MBRL is nicely outlined and the idea behind the proposed approach is explained intuitively.
2. Subsections 5.3 and 5.4 show promising results (but would still like to see more extensive results).
3. Subsection 5.5 discovers an important limitation of ATC which might inspire new research directions.

**Weaknesses:**

1. The greatest weakness of this work is the lack of novelty. It is a direct application of ATC by [Stooke et al. 2021](https://arxiv.org/pdf/2009.08319) to DreamerV3 [Hafner et al. 2024](https://arxiv.org/pdf/2301.04104v2). While this is a clear weakness, it can still be executed in such a way that it is more fruitful for the research community. One option would be to explore the idea of ATC to the larger scope of MBRL, not just Dreamer. Another option that would compensate for this weakness would be extensive empirical experiments and novel findings.
    * Note that the lack of algorithmic or theoretical contributions means that I am evaluating your work mostly on the empirical results presented.
2. I generally found Section 4 to be poorly written and I struggled to understand this relatively simple method.
    * I found the introduction of the encoders and variables produced (as outputs of models) highly confusing. Section 4.1 introduces $x_t$ which is not part of Dreamer but never explains what produces this variable. Then seemingly all the model components are presented in the equations of Section 4.1, yet it still unclear where this is coming from. Section 4.2 then adds an additional deterministic decoder component which produces $x_t$ forcing readers to circle back and re-read the whole section to understand the flow of variables. While I am not sure what is the best idea, I can suggest (1) re-writing the whole section as if you are explaining Dreamer from scratch or (2) assuming dreamer formulation and tightly focusing on how you modify the training of the encoder.
    * It is not clear from the text along what the transformation matrix W is. Given that this is your core algorithmic contribution, I strongly suggest expanding on this.
3. Mathematical formulations are sloppy.
    * Equations are not numbered.
    * Variables are used liberally. Main loss equation (right above subsection 4.3), you use T very liberally. Following your problem definition, T should be episode length. If so, I doubt you are computing this loss over the whole episode. Furthermore, the expectation doesn’t seem right as there is a replay buffer here.
4. Results are a combination of limited and unconvincing.
    * In my opinion, when you add complexity to an already very complicated algorithm such as Dreamer, the benefits of this have to be very clearly visible. This is the lens from which I am approaching this.
    * Subsection 5.1 attempts to answer “Does our method learn useful endogenous representation for the world model?” by using one subjective demonstration on a single task with a single seed. I find this insufficiently rigorous do draw any conclusions from. I suggest attempting to answer this question by focusing on specific tasks that have these types of environment dynamics and presenting numerical results from that.
    * Subsection 5.2 shows promising results but I find it very limited as it only shows 3 environments. I believe we are far beyond this point as a research community. Furthermore, if you want to showcase sample efficiency, I suggest showing return vs samples in Figure 3.
    * Table 1 is difficult to read and draw conclusions from as noted by [Agarwal et al., 2022](https://arxiv.org/abs/2108.13264). As you already have this, I would strongly suggest replacing Table 1 with Figure 6.
5. The proposed approach also includes an image augmentation technique in the form of a shift operation. It is unclear how much this helps to achieve the final results. You should include an ablation of removing this but keeping the ATC loss.

**Questions:**

1. What are the evaluation metrics used in Figure 3? They are not described in the text up to that point.
2. One of the interesting things about your method is that it allows for training MBRL algorithms without reconstruction. Why didn't you explore other world model frameworks such as TDMPC [(Hensen et al, 2022)](https://arxiv.org/abs/2203.04955)?
3. Do you have any intuition as to why your proposed method doesn't work as well on the Distracting Control Suite?

---

> ### Author Response · Authors · 2024-12-03
> **W1 and W2**
>
> W1. The greatest weakness of this work is the lack of novelty. It is a direct application of ATC by Stooke et al. 2021 to DreamerV3 Hafner et al. 2024. While this is a clear weakness, it can still be executed in such a way that it is more fruitful for the research community. One option would be to explore the idea of ATC to the larger scope of MBRL, not just Dreamer. Another option that would compensate for this weakness would be extensive empirical experiments and novel findings.
>
> Note that the lack of algorithmic or theoretical contributions means that I am evaluating your work mostly on the empirical results presented.
>
> Q1. We thank the reviewer for the insightful feedback. Indeed, we will work on that in a later iteration of our work.
>
> W2. I generally found Section 4 to be poorly written and I struggled to understand this relatively simple method.
>
>  -  I found the introduction of the encoders and variables produced (as outputs of models) highly confusing. Section 4.1 introduces $x_t$ which is not part of Dreamer but never explains what produces this variable. Then seemingly all the model components are presented in the equations of Section 4.1, yet it still unclear where this is coming from. Section 4.2 then adds an additional deterministic decoder component which produces $x_t$ forcing readers to circle back and re-read the whole section to understand the flow of variables. While I am not sure what is the best idea, I can suggest (1) re-writing the whole section as if you are explaining Dreamer from scratch or (2) assuming dreamer formulation and tightly focusing on how you modify the training of the encoder.
> - It is not clear from the text along what the transformation matrix W is. Given that this is your core algorithmic contribution, I strongly suggest expanding on this.
>
> A2. We thank the reviewer for the feedback. To address it, we have made a major revision of our section 4. It will be great for our work if the reviewer could read our current draft and tell us if that makes the reading clearer.

---

> ### Author Response · Authors · 2024-12-04
> **W3 and W4**
>
> W3. Mathematical formulations are sloppy. Equations are not numbered; Variables are used liberally. Main loss equation (right above subsection 4.3), you use T very liberally. Following your problem definition, T should be episode length. If so, I doubt you are computing this loss over the whole episode. Furthermore, the expectation doesn’t seem right as there is a replay buffer here.
>
> A3. Thank you for the comments. We have numbered the equations and significantly updated Section 4 to use notations in a clearer manner. We have made it clear in our manuscript that $T$ denotes the end point of the positive trajectory, which is consistent with the previous literatures.
>
> Regarding the expectation, we followed the expectation equation from the DreamerV3 paper.
>
> We hope our revisions address the reviewer's concerns.
>
> W4.1. In my opinion, when you add complexity to an already very complicated algorithm such as Dreamer, the benefits of this have to be very clearly visible. This is the lens from which I am approaching this.
>
> Subsection 5.1 attempts to answer “Does our method learn useful endogenous representation for the world model?” by using one subjective demonstration on a single task with a single seed. I find this insufficiently rigorous do draw any conclusions from. I suggest attempting to answer this question by focusing on specific tasks that have these types of environment dynamics and presenting numerical results from that.
>
> A4.1. We do not think our method is complex, as it is a combination of two popular methods. We keep it simple to evaluate our central question of evaluating the benefit of latent prediction for MBRL algorithms such as DreamerV3. Our analysis on the Atari100k benchmark shows that our method improves DreamerV3. Indeed it is also well motivated by mental simulation of animal brains [1].
>
> Besides, the Authors would argue that this section 5.1 has a place in this paper, as it shows how the problem with the representation of DreamerV3 that we highlighted in the introduction is truly an issue, and that we managed to tackle it. Zooming out and reflecting on our initial example of the problem statement indeed has a place in the beginning of the section that we present our results in.
>
> So, although it is only one environment, it concludes that we learned to represent the ball—minor endogenous information—we speak so highly about in the beginning.
>
> Unfortunately, due to computational and time limitations we were unable to run new experiments in order to present the numerical results suggested. However, the Authors thank the reviewer for this comment—the suggested approach would be convincing as an additional subsection.
>
>
> Q4.2. Subsection 5.2 shows promising results but I find it very limited as it only shows 3 environments. I believe we are far beyond this point as a research community. Furthermore, if you want to showcase sample efficiency, I suggest showing return vs samples in Figure 3.
>
> Table 1 is difficult to read and draw conclusions from as noted by Agarwal et al., 2022. As you already have this, I would strongly suggest replacing Table 1 with Figure 6.
>
> A4.2. To address the reviewer's comment, we have moved Table 1 to the Appendix and put Figure 6 to its place as suggested. Due to time limitations we left Figure 3 as it was.
>
> That said,  in the additional ablation experiment (Section 5.6) we ran experiments on more environments and invite the review to our revision.
>
> 1. Nayebi, Aran, et al. "Neural foundations of mental simulation: Future prediction of latent representations on dynamic scenes." Advances in Neural Information Processing Systems 36 (2024).

---

> ### Author Response · Authors · 2024-12-04
> **W5**
>
> W5. The proposed approach also includes an image augmentation technique in the form of a shift operation. It is unclear how much this helps to achieve the final results. You should include an ablation of removing this but keeping the ATC loss.
>
> A5. We thank the reviewer for such insightful question. We agree with the reviewer and we have worked on an ablation study that assesses the contribution of the following components:
>
> Data augmentation with random shifts: We remove the ATC loss and add data augmentation to the training trajectories for the DreamerV3 world model loss. Specially, we take the observations, add random shifts to them, and use these observations to learn a world model. Indeed, these augmentations only happen during the training and not during evaluations.
>
> Augmented temporal contrast loss: We remove the data augmentations that was being applied to anchor and positive. Afterwards, we calculate the ATC loss.
>
> We compare these two scenarios with our method to understand the exact contribution of each component to the reported performance improvements.
>
> We validate the ablation on 5 Atari games (Alien, Asterix, Krull, Breakout, MsPacman) over 5 seeds. We report the results of the ablations in Section 5.6.
>
> Figure 7 illustrates that the removal of the ATC loss results in poorer performance, specially in Breakout and Asterix. Additionally, leveraging the shift augmentation has a steady impact in all the games tested. However, their influence appears negligible in Krull, which is consistent with our findings in Section 5.5 where we show it is improved with higher values of .
>
> Please let us know if you have any remaining doubts or concerns.

---

> ### Author Response · Authors · 2024-12-04
> **Q1-3**
>
> Q1. What are the evaluation metrics used in Figure 3? They are not described in the text up to that point.
>
> A. Figure 3 (that is Figure 4 in our updated manuscript) compares the average return of our method and DreamerV3. We averaged the average returns across different random seeds to present the numbers on the y-axis.
>
>
> Q2. One of the interesting things about your method is that it allows for training MBRL algorithms without reconstruction. Why didn't you explore other world model frameworks such as TDMPC (Hensen et al, 2022)?
>
> A. It does not allowed training MBRL algorithms without reconstruction. We have added ATC loss to evaluate our central question "Does forward prediction help learn better representations for MBRL?".
>
> Indeed, it would help to test on more tasks and methods of our validity and our future iterations will address it.
>
> Q3. Do you have any intuition as to why your proposed method doesn't work as well on the Distracting Control Suite?
>
> A. It is due to the fact that our method only identifies endogenous information by "moving objects", i.e., features that are important for dynamics prediction. DCS makes identifying  endogenous information harder by having unrelated dynamics backgrounds. This is why our method does not work as well.
>
> That said, Our method does not improve DreamerV3 on the easy tasks only. We attribute this to the fact that the easy samples contain so few visual distractions that it is too easy for DreamerV3, even with its poorer representation. As the task gets harder, DreamerV3 will start to fail due to its poor representation, as opposed to our method that outperforms it with its better representation.

---

### Official Review · Reviewer_K1SW · 2024-11-01

**Soundness:** 2
**Presentation:** 1
**Contribution:** 3
**Rating:** 3
**Confidence:** 4

**Summary:**

This paper introduces a new world model for deep reinforcement learning by integrating DreamerV3 with augmented temporal contrast (ATC) as an auxiliary loss. This added component enhances the world model's robustness to exogenous, irrelevant features. The authors evaluate their approach on the Atari 100k benchmark and conduct several ablation studies.

**Strengths:**

- [S1] The paper proposes a novel and meaningful combination of established methods, leveraging ATC to strengthen DreamerV3's auxiliary loss, which is a promising direction.
- [S2] The paper demonstrates a good selection of experiments, yielding promising—albeit preliminary—results on the Atari 100k benchmark.

**Weaknesses:**

- [W1] Figure 2, in its current form, is not particularly convincing. Firstly, it's unclear whether it shows reconstructions from a real trajectory or a rollout generated by the world model. This distinction should be clarified both in the figure caption and in Section 5.1. I assume it represents a rollout, as the trajectories appear noticeably different. For instance, the second row might depict the initial situation in Breakout, where the paddle moves before the 'FIRE' command, making it reasonable to model the absence of the ball. To strengthen this experiment, the authors should consider using consistent starting states for rollouts, especially since the evaluation relies exclusively on qualitative assessment without quantitative metrics.
- [W2] The "Implementation" section in 4.2 is confusing. The variable $\overline{x}$ (presumably the latent features from the target encoder) is used without prior introduction. The terms "forward propagation $p_t$" (line 248) need clarification—do the authors mean "forward prediction"? Additionally, the notation $p_t = x_t + h_\phi(x_t)$ suggests a residual connection, which requires further explanation. Lastly, it's unclear how the matrix $W$ is updated; if it's part of $\phi$, this should be made explicit.
- [W3] The paper does not sufficiently differentiate which components are inherited from ATC and which are original contributions. For example, Section 4.2 introduces anchors and positives from ATC, but it is unclear whether elements like the InfoNCE loss, target encoder with EMA, and data augmentation are standard in ATC or were newly introduced.
Additionally, it would be beneficial to reference Stooke et al. (2021) in lines 170-176.
- [W4] All experiments use only three random seeds, and ablation studies are limited to three games, reducing the generalizability of the findings. Although the authors acknowledge this limitation, and I understand that expanding these experiments may be challenging, doing so would significantly strengthen the paper's significance.
- [W5] More presentation issues:
  - Equations should be numbered for ease of reference.
  - The encoder notation $g_\phi$ appears in Section 4.2 without prior introduction. Including it in the overview on lines 189-197 would improve clarity.
  - Section 7 could be merged with Section 6 for a more cohesive structure.
  - Moving Section 5.4 to the beginning of Section 5 would clarify the experimental setup. Additionally, including a description of the Atari 100k benchmark in this section (noting it comprises a subset of 26 games) would be helpful.
  - Lines 207-209: Consider rephrasing for clarity: "For regularization, we use a target encoder updated via exponential moving averages (EMA) and apply data augmentation to the sensory observations."
  - Figure 2 could be improved by reducing font size, labeling rows, and adding whitespace between them.
  - Line 125: "scricter" should be "stricter"
  - Line 186: Correct symbols to $\hat{r}_t$ and $\hat{o}_t$.

The paper has substantial potential, presenting a valuable new approach; however, it currently feels incomplete. Addressing the issues outlined above would strengthen its contribution, clarity, and rigor.

**Questions:**

- [Q1] Why is an additional recurrent predictor used? Could the RSSM not serve this purpose?
- [Q2] I assume a frame skip of 4 is applied, meaning the prediction horizon of $K = 4$ would cover 16 frames. Could the authors confirm?

---

> ### Author Response · Authors · 2024-12-02
> **W1 and W2**
>
> W1. Figure 2, in its current form, is not particularly convincing. Firstly, it's unclear whether it shows reconstructions from a real trajectory or a rollout generated by the world model. This distinction should be clarified both in the figure caption and in Section 5.1. I assume it represents a rollout, as the trajectories appear noticeably different. For instance, the second row might depict the initial situation in Breakout, where the paddle moves before the 'FIRE' command, making it reasonable to model the absence of the ball. To strengthen this experiment, the authors should consider using consistent starting states for rollouts, especially since the evaluation relies exclusively on qualitative assessment without quantitative metrics.
>
> A1. Thank you for your insightful feedback, you are correct that the figure lacked clarity regarding whether it displays reconstructions from real trajectories or rollouts. We have updated both the figure caption and Section 5.1 to state that the images represent rollouts generated by the world model.
>
> For the updated Figure 2 we have used consistent starting states across all rollouts at 100k steps, providing a more accurate and convincing comparison. Due to limited time, we were unable to include 40k and 60k steps with consistent starting states. However, we believe that the updated 100k step rollouts offer a clearer demonstration of the model's performance and address the core of your concerns.
>
>
> W2. The "Implementation" section in 4.2
>
> A2. We updated our Implementation section significantly to address all of the reviewers' concerns.
>
> We elaborated on the residual connection in the recurrent predictor and corrected the notations to be clearer. “Forward propagation” was indeed meant to be “forward prediction”, thank you for bringing this to our attention. Now we also explain how the contrastive transformation matrix W is updated: we calculate the differences between the logits and true labels using a cross-entropy loss.

---

> ### Author Response · Authors · 2024-12-02
> **W3, W4 and W5**
>
> W3.  The paper does not sufficiently differentiate which components are inherited from ATC and which are original contributions. For example, Section 4.2 introduces anchors and positives from ATC, but it is unclear whether elements like the InfoNCE loss, target encoder with EMA, and data augmentation are standard in ATC or were newly introduced. Additionally, it would be beneficial to reference Stooke et al. (2021) in lines 170-176.
>
> A3. We added an extra motivational section to Section 4.2 and made explicit that we incorporated ATC as it is to the DreamerV3 architecture for the reasons explained below. We also referenced Stooke et al. (2021) as suggested.
>
> Our work is inspired by animal cognitive process, specifically from this work Nayebi et al. (2023) where the authors show that animal brains' capability of processing important information mimics more closely to predicting futures in the latent representation.
> The authors report, "...future prediction within the latent space of a video foundation model, pretrained on diverse egocentric sources, aligns best with high-throughput neural and behavioral patterns in scenes that it was not originally trained on. Our findings indicate that primate mental simulation harbors robust inductive biases, and is so far most consistent with predicting the future state of its environment in a latent space that is reusable across dynamic environments."
>
> As our work also incorporates the same pathway, the ATC loss was suitable as it was to help the model to focus on features that are useful for predicting the dynamics.
>
> W4. All experiments use only three random seeds, and ablation studies are limited to three games, reducing the generalizability of the findings. Although the authors acknowledge this limitation, and I understand that expanding these experiments may be challenging, doing so would significantly strengthen the paper's significance.
>
> A4. Unfortunately, due to computational and time limitations we were unable to conduct experiments with more random seed. However, we updated our manuscript (section 5.6) with new ablation studies on ATC loss and shift augmentation on five games, showing that our final method with ATC loss and shift augmentation combined performs the best overall.
>
> W5. More presentation issues.
>
> A5. We thank the reviewer for noticing these problematic parts. These have been fixed in our updated manuscript.

---

> > ### Author Response · Authors · 2024-12-03
> > **Q1 and Q2**
> >
> > Q1. Why is an additional recurrent predictor used? Could the RSSM not serve this purpose?
> >
> > A1. To make the implementation as simple as possible, we utilized the architecture utilized by Stooke et al. (2021). Indeed, as the reviewer mentions, the RSSM could be utilized, but it would make the connections complicated.
> >
> > Q2. I assume a frame skip of 4 is applied, meaning the prediction horizon of $K = 4$ would cover 16 frames. Could the authors confirm?
> >
> > A2. This is correct. We follow the previous methods and apply a frameskip of 4, so $K=4$ indeed covers 16 frames.

---

### Official Review · Reviewer_VEKG · 2024-11-02

**Soundness:** 2
**Presentation:** 2
**Contribution:** 2
**Rating:** 3
**Confidence:** 4

**Summary:**

This paper introduces a model-based reinforcement learning method designed to guide the world model towards crucial task-related information. It extends the DreamerV3 framework and integrates an augmented temporal contrast (ATC) loss as an auxiliary loss. The effectiveness of this approach is evaluated on the Atari and DeepMind Control Suite with distractions, demonstrating comparable performance to baseline methods.

**Strengths:**

1. This paper is relatively straightforward and easy to comprehend.
2. It learns a robust world model to focus on the task-relevant information.

**Weaknesses:**

1. This paper primarily focuses on endogenous information that directly influences the success of the agent's strategy. However, when using reconstruction losses, the model will inevitably consider exogenous information, and it may be better to replace the reconstruction loss with other loss functions, such as DreamerPro [1]. Additionally, the process by which the augmented temporal contrast loss guides the model to focus on relevant information remains unclear.
2. In Lines 205-206, the predictor model is employed to forecast the positive embeddings $(x_{t+K}, x_{t+1+K}, \ldots, x_{t+T+K})$ based on the anchor embeddings $(x_t, x_{t+1}, \ldots, x_{t+T})$. Does this suggest that $x_t$ is leveraged to predict $x_{t+K}$, and what actions can be taken to implement the transition from the time step $t$ to time step $t+K$?
3. Several previous works have tackled visual distractions to enhance robustness in representation learning, including DreamerPro [1], Iso-Dream [2], DBC [3], and Denoised-MDP [4]. It is better to compare these method with the proposed method.
4. The current experimental environment may be somewhat limited and simplistic for effectively showcasing the proposed method's effectiveness. Considering a more realistic environment like CARLA, which incorporates a range of distractions both related and unrelated to the agent's actions, could provide valuable insights.
5. In this paper, the authors did not analyze whether the data augmentation or the augmented temporal contrast loss had a greater impact on the model performance. It would be beneficial to conduct ablation studies to explore this further.
6. In Fig.1, the arrow between $h_t$ and $z_{t+1}$ is incorrect.

[1] DreamerPro: Reconstruction-free model-based reinforcement learning with prototypical representations.

[2] Model-Based reinforcement learning with isolated imaginations.

[3] Learning invariant representations for reinforcement learning without reconstruction.

[4] Denoised mdps: Learning world models better than the world itself.

**Questions:**

Please see the weaknesses section.

---

> ### Author Response · Authors · 2024-11-28
> **Q1 and Q2**
>
> Q1. This paper primarily focuses on endogenous information that directly influences the success of the agent's strategy. However, when using reconstruction losses, the model will inevitably consider exogenous information, and it may be better to replace the reconstruction loss with other loss functions, such as DreamerPro [1]. Additionally, the process by which the augmented temporal contrast loss guides the model to focus on relevant information remains unclear.
>
> A1. We thank the reviewer for asking such insightful question. We will try to answer as clearly as possible, including our motivation and the insights.
>
> Our work is inspired by animal conginitive process, specifically from this work [1] where the authors show that animal brains' capability of processing important information mimics more closely to predicting futures in the latent representation.
>
> The authors report, "...future prediction within the latent space of a video foundation model, pretrained on diverse egocentric sources, aligns best with high-throughput neural and behavioral patterns in scenes that it was not originally trained on. Our findings indicate that primate mental simulation harbors robust inductive biases, and is so far most consistent with predicting the future state of its environment in a latent space that is reusable across dynamic environments."
>
> Our work also incorporates the same pathway. The augmented temporal contrast loss focuses on features that are useful for predicting the dynamics, which is mostly the cases with endogenous features.
>
> Motivated by [1], we note in our introduction "Can a forward-predictive scheme in the joint embedding space help to learn a better representation for training world models?". Thus, we decide to make as little as possible to the DreamerV3 to evaluate the findings in the case of world models. This is why we do not consider DreamerPro loss instead of the reconstruction, although we agree with the reviewer that it is better to replace the reconstruction loss with other loss functions.
>
> Q2. In Lines 205-206, the predictor model is employed to forecast the positive embeddings $(x_{t+K}, x_{t+1+K}, \dots, x_{t+T+K})$  based on the anchor embeddings $(x_{t}, x_{t+1}, \dots, x_{t+T})$ . Does this suggest that $x_t$ is leveraged to predict $x_{t+K}$, and what actions can be taken to implement the transition from the time step $t$ to time step $t+K$?
>
> A2. We thank the reviewer for such detailed-oriented question. While the reviewer is correct that $x_t$ is leveraged to predict $x_{t+K} and so on, the use of a contrastive loss on the learnt logits ensures that the model forced to learn the associate the order of the paired anchor and positives.
>
> For example, $x_t$ and $x_{t+K}$ should be associated with the position 0, $x_{t+1}$ and $x_{t+K+1}$ should be associated with the position 1, and so on.
>
> Indeed, our method relies on the fact that by increasing the information content that helps with associating the positives with the anchor (in correct order) in the representation, it should learn features that are more useful for predicting the dynamics.
>
> 1. Nayebi, Aran, et al. "Neural foundations of mental simulation: Future prediction of latent representations on dynamic scenes." Advances in Neural Information Processing Systems 36 (2024).

---

> ### Author Response · Authors · 2024-11-28
> **Q3 and Q4**
>
> Q3. Several previous works have tackled visual distractions to enhance robustness in representation learning, including DreamerPro [1], Iso-Dream [2], DBC [3], and Denoised-MDP [4]. It is better to compare these method with the proposed method.
>
> A3. As described in Q1, one of our key motivations for our work is to show with that a simple change to the base method, specifically, forward-predictive scheme in the joint embedding space, can improve the base method's robustness.
>
> Indeed, we do not claim to tackle the robustness problem, rather show that our augmentation helps with the robustness.
>
> Besides, while the experiments would be an interesting addition to our current work, the limited time period in author-reviewer discussion phase does not permit us to carry through the experiments. These being the case, we do not think the experiments are essential at this stage.
>
> Q4. The current experimental environment may be somewhat limited and simplistic for effectively showcasing the proposed method's effectiveness. Considering a more realistic environment like CARLA, which incorporates a range of distractions both related and unrelated to the agent's actions, could provide valuable insights.
>
> A4. We agree with the reviewer that a more realistic environment like CARLA can provide valuable insights and thank the reviewer for this direction. However, due to resource constraints, we could not perform the experiment.

---

> ### Author Response · Authors · 2024-11-28
> **Q5 and Q6**
>
> Q5. In this paper, the authors did not analyze whether the data augmentation or the augmented temporal contrast loss had a greater impact on the model performance. It would be beneficial to conduct ablation studies to explore this further.
>
> A5. We thank the reviewer for such insightful question. We agree with the reviewer and we have worked on an ablation study that assesses the contribution of the following components:
>
> Data augmentation with random shifts: We remove the ATC loss and add data augmentation to the training trajectories for the DreamerV3 world model loss. Specially, we take the observations, add random shifts to them, and use these observations to learn a world model. Indeed, these augmentations only happen during the training and not during evaluations.
>
> Augmented temporal contrast loss: We remove the data augmentations that was being applied to anchor and positive. Afterwards, we calculate the ATC loss.
>
> We compare these two scenarios with our method to understand the exact contribution of each component to the reported performance improvements.
>
> We validate the ablation on 5 Atari games (Alien, Asterix, Krull, Breakout, MsPacman) over 5 seeds. We report the results of the ablations in Section 5.6.
>
> Figure 7 illustrates that the removal of the ATC loss results in poorer performance, specially in Breakout and Asterix. Additionally, leveraging the shift augmentation has a steady impact in all the games tested. However, their influence appears negligible in Krull, which is consistent with our findings in Section 5.5 where we show it is improved with higher values of $K$.
>
> Please let us know if you have any remaining doubts or concerns.
>
> Q6. In Fig.1, the arrow between $h_t$  and $z_{t+1} is incorrect.
>
> A6. We thank the reviewer for noticing this. This has been fixed in our updated manuscript.

---

### Meta-Review · Area_Chair_YiMM · 2024-12-23

**Metareview:**

This paper introduces an enhancement to model-based RL methods like DreamerV3, by adding a temporal loss function in the joint embedding space. This is meant to focus the model to concentrate on task specific features, to improve robustness and sample efficiency.

The strength of the paper is the introduction of ATC to address a limitations of pixel level reconstruction in MBRL. But there was consensus about the limited novelty during the review process. While the work is methodologically sound and well-motivated, it does not introduce significant theoretical contributions or demonstrate sufficient empirical generalization across diverse tasks and frameworks.

**Additional Comments On Reviewer Discussion:**

After the rebuttal process, concerns about limited novelty, and constrained experimental and base scope led to a recommendation of rejection from authors at this stage.

---

### Decision · Program_Chairs · 2025-01-22

Reject